# Spatio-Temporal Vegetation Dynamic and Persistence under Climatic and Anthropogenic Factors

**Barjeece Bashir** [1,2], **Chunxiang Cao** [1,2,*], **Shahid Naeem** [2,3], **Mehdi Zamani Joharestani** [1,2], **Xie Bo** [1,2], **Huma Afzal** [4], **Kashif Jamal** [2,5] **and Faisal Mumtaz** [1]

1   State Key Laboratory of Remote Sensing Science, Aerospace Information Research Institute, Chinese Academy of Sciences, Beijing 100101, China; barjeece@radi.ac.cn (B.B.); madiz@radi.ac.cn (M.Z.J.); xiebo@radi.ac.cn (X.B.); faisal@radi.ac.cn (F.M.)
2   University of Chinese Academy of Sciences, Beijing 100094, China; shahidn@igsnrr.ac.cn (S.N.); kashif@lzb.ac.cn (K.J.)
3   Institute of Geographical Sciences and Natural Resources Research (IGSNRR), Beijing 100101, China; humaafzal51@gmail.com
4   Department of Geography, Government Degree College for Women Sahianwala, Faisalabad 37701, Pakistan
5   Key Laboratory of Remote Sensing and Geospatial Science, Northwest Institute of Eco-Environment and Resources, Chinese Academy of Sciences, Lanzhou 730000, China
*   Correspondence: caocx@radi.ac.cn; Tel.: +86-139-1161-0226

**Abstract:** Land degradation reflected by vegetation is a commonly used practice to monitor desertification. To retrieve important information for ecosystem management accurate assessment of desertification is necessary. The major factors that drive vegetation dynamics in arid and semi-arid regions are climate and anthropogenic activities. Progression of desertification is expected to exacerbate under future climate change scenarios, through precipitation variability, increased drought frequency and persistence of dry conditions. This study examined spatiotemporal vegetation dynamics in arid regions of Sindh, Pakistan, using annual and growing season Normalized Difference Vegetation Index (NDVI) data from 2000 to 2017, and explored the climatic and anthropogenic effects on vegetation. Results showed an overall upward trend (annual 86.71% and growing season 82.7%) and partial downward trend (annual 13.28% and growing season 17.3%) in the study area. NDVI showed the highest significant increase in cropland region during annual, whereas during growing season the highest significant increase was observed in savannas. Overall high consistency in future vegetation trends in arid regions of Sindh province is observed. Stable and steady development region (annual 48.45% and growing 42.80%) dominates the future vegetation trends. Based on the Hurst exponent and vegetation dynamics of the past, improvement in vegetation cover is predicted for a large area (annual 44.49% and growing 30.77%), and a small area is predicted to have decline in vegetation activity (annual 0.09% and growing 3.04%). Results revealed that vegetation growth in the study area is a combined result of climatic and anthropogenic factors; however, in the future multi-controls are expected to have a slightly larger impact on annual positive development than climate whereas positive development in growing season is more likely to continue in future under the control of climate variability.

**Keywords:** vegetation dynamic trend; Hurst exponent; MODIS; climate variability; remote sensing

## 1. Introduction

Desertification is a form of land degradation, due to which a relatively dry region typically loses its water bodies, as well as vegetation and wildlife [1]. In arid, semi-arid, and dry sub-humid regions, desertification is caused by a combination of different climatic and anthropogenic factors [2].

These regions provide food to more than 40% of the world's population [3] and are inhabited by more than 2 billion people [4]. Because of its impacts on environmental quality (dust storms, emission of trace gases, soil erosion, etc.) and human populations (sustainability, food security, economics, etc.) land degradation is a vital societal concern [5]. Dearth of understanding about the desertification rate and its extent in dry regions of the world have been described as one of the key issues of our time [6].

Previous studies have used land degradation reflected by vegetation to monitor desertification [7–9] as vegetation cover is a highly unstable and delicate portion of the ecosystem [10]. Due to the biophysical response of plant respiration, evapotranspiration, and photosynthesis, vegetation has an obvious relationship with climatic factors [11–14]. Several studies revealed that the main climatic factors influencing vegetation production and activity in arid and semi-arid regions are temperature and rainfall [15] but the most significant and robust limiting factor of vegetation growth is precipitation [16]. Soil surface exposed to wind and water erosion causes losses of soil nutrients that limit vegetation growth. Progression of desertification is expected to exacerbate under future climate change scenarios, through precipitation variability, increased drought frequency and persistence of dry conditions [17]. Climate has been warming since 1950 in almost all the regions of the world including Asia. Precipitation is decreasing significantly as pointed out in the fifth assessment report of the Intergovernmental Panel on Climate Change (IPCC) [18].

For the past three decades, remotely sensed data were widely used for the monitoring of ecosystems and hazards like desertification [2]. The development of remote sensing tools during the last two decades provides an opportunity to assess desertification in arid and semi-arid regions at a reasonable spatial scale [19]. Remote sensing satellite images from Sentinel-2, Moderate Resolution Imaging Spectroradiometer (MODIS) and Landsat were used by some researchers to monitor and map desertification-sensitive areas at different scales [16,20,21]. However, some researchers have also analyzed the persistence of land degradation or restoration based on vegetation trends, finding results that can provide information about long-term memory of degradation or restoration after the analysis period and availability of data [22–24].

To detect the vegetation dynamics and its response to climatic factors, one of the most commonly used method is to use satellite derived Normalized Difference Vegetation Index (NDVI) [25,26] because NDVI is highly correlated with leaf area, potential photosynthesis of vegetation, photosynthetically active biomass, and chlorophyll abundance [15,27]. The NDVI, quantifies vegetation by measuring the difference between infrared and near infrared channel remote sensing data. NDVI is linearly related to vegetation distribution density [25,28]. Despite the fact that NDVI has several limitations, including noisy canopy background signal in the regions of sparse vegetation and saturation in highly vegetated canopy, this index is largely used from local to global scale in vegetation dynamics studies [16,22] due to its simplicity and robustness [23]. Since the launch of the Moderate Resolution Imaging Spectroradiometer (MODIS) data have been considered as state-of-the-art. With the large temporal resolution MODIS NDVI products are largely been used for vegetation research [22,29].

Desertification is a global environmental problem, but the situation is more critical in Pakistan. Eighty percent of Pakistan's land is arid and semi-arid, and almost three-quarters of the country's land is either already affected by desertification or likely to be affected by it [30]. A chain of interrelated environmental, social, and economic issues associated with land degradation has been generated by population pressure along with increasing demand for food, fuel, and fodder in Pakistan [31]. For the past two decades, the Sindh province of Pakistan has been prone to harsh drought [32]. District Sanghar is among the highly vulnerable districts of Sindh province that are facing droughts [33]. The main characteristics of Sindh include water scarcity [34], droughts [33], increasing livestock [35] and inadequate human activities. These characteristics can limit vegetation growth; thereby, droughts may exaggerate the desertification rate if carrying capacity continues to decrease. The government has implemented several ecological restoration projects to address the devastating land desertification [36,37]. To assess the outcomes and persistence of the government's policies it is essential to monitor vegetation dynamic activity.

The rescaled range analysis (R/S analysis) Hurst exponent, based on auto-covariance (a function of lag [38] that gives the covariance of the process with itself at pairs of time points), can be applied as a parameter to spot the consistency of time series data of a natural phenomenon [22]. This parameter has widely been used in several fields like geology [39], hydrology [40], economics [41], and climatology [42]. The auto-covariance function can capture long-term memory effects that decay exponentially with a spectral density that tends to infinity [22].

To the author's best knowledge, previous researchers did not discuss the persistence of future vegetation trends under different deriving forces. The present study can help to understand the positive and negative impacts of desertification drivers more clearly and to build better policies for the future. The specific aims of this study are to (1) monitor spatio-temporal vegetation trends from 2000 to 2017 and mapping areas with significant vegetation change, (2) distinguish the influence of anthropogenic and climatic factors on positive and negative vegetation trends, and (3) investigate the influence of anthropogenic and climatic drivers on the persistence of future vegetation. The findings of this study will provide a baseline reference to policy-makers to take informed decisions in the complex landscape regions for the vegetation conservation and restoration.

## 2. Materials and Methods

### 2.1. Study Area

The study area is the arid region of Sindh Province in southeast Pakistan (Figure 1). The area has total coverage of 98,107.9 km$^2$. Based on Thornthwaite classification, Sindh is divided into semiarid tropical types of climate which covers lower Sindh and arid tropical climate of upper Sindh [43]. According to the International Geosphere-Biosphere program's (IGBP) type 1 MODIS land cover classification of study area, barren land covers 37.4%, croplands account for 36.1%, grassland covers 13%, shrublands covers 11.7%, water bodies cover 0.5%, urban accounts for 0.3%, and savannas cover 0.2% of the total area (Figure 2). The whole province receives a low amount of rain and has a high temperature. The mean annual average air temperature varies between 22 to 28 °C, the mean growing season air temperature varies between 28 to 34 °C, and the mean annual precipitation varies between 120 mm to 285 mm, and mean growing season precipitation varies between 86 to 220 mm. Figure 3 illustrates the difference between growing season (GS) and annual climatic factors.

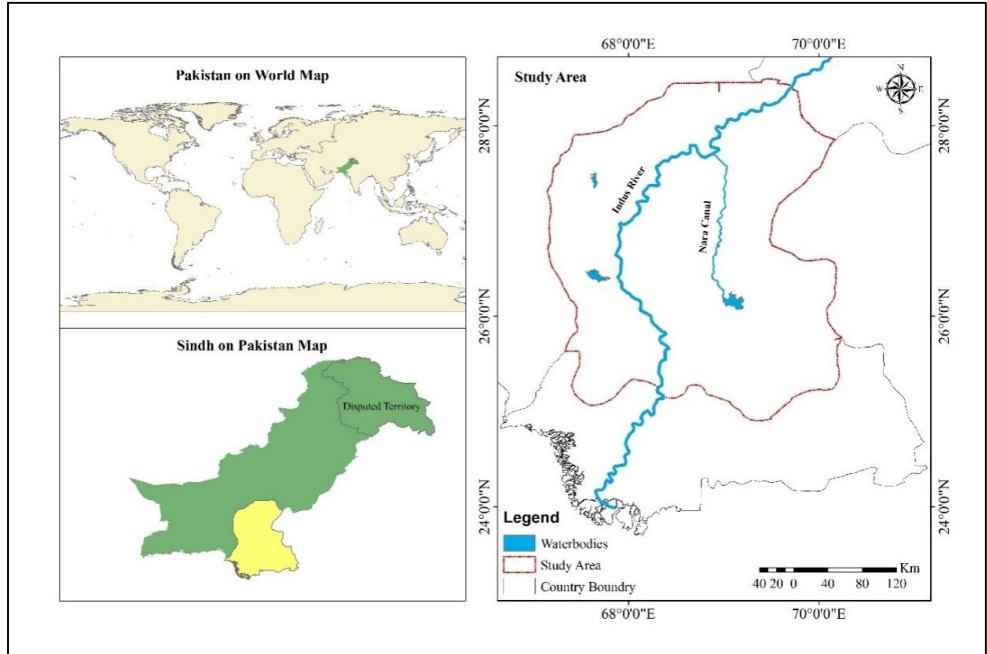

**Figure 1.** Location of study area.

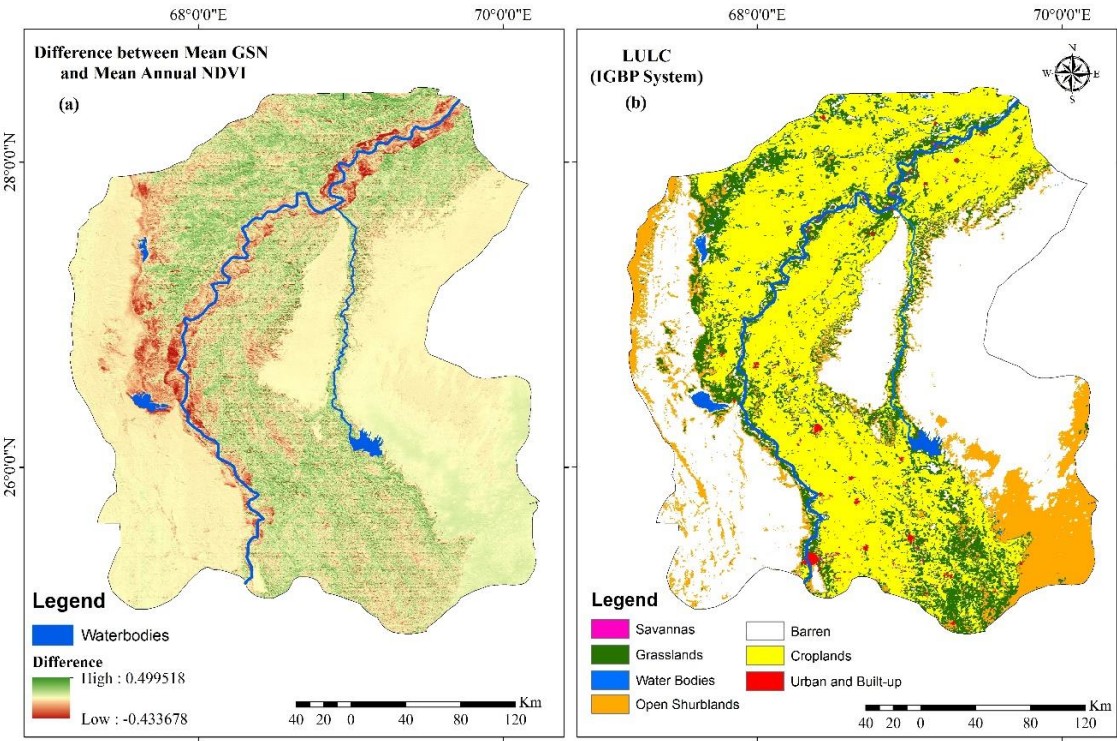

**Figure 2.** (**a**) Difference between Mean growing season (GS) and Mean Annual Normalized Difference Vegetation Index (NDVI) and (**b**) land cover map of arid regions of Sindh.

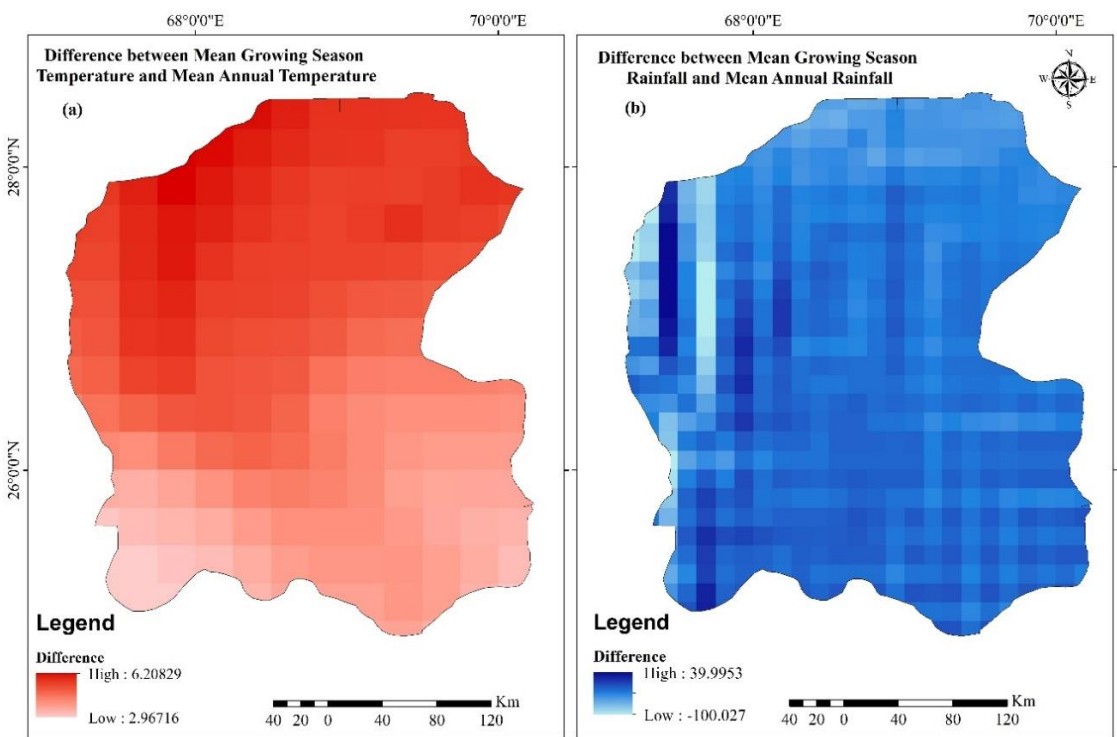

**Figure 3.** (**a**) Difference between mean Growing Season Temperature and Mean Annual Temperature; (**b**) Difference between mean Growing season Rainfall and Mean Annual Rainfall.

*2.2. Data*

Eighteen years of MODIS (MOD13Q1) NDVI data from 2000 to 2017 with a 16 day temporal resolution is used in this study. These images are calculated from daily, atmospherically corrected, surface reflectance and have 250 m spatial resolution. They have been masked by clouds, water, heavy aerosols, and cloud shadow and pixels, with the maximum quality were selected for each 16-day composite [44]. To obtain annual and growing season NDVI (May to November) for each year (2000 to 2017), the NDVI values were averaged. European Centre for Medium-Range Weather Forecasts (ECMWF) reanalysis data is valuable atmospheric data for climatic and weather research [45]. The monthly gridded time-series data ERA5, reanalysis precipitation, and air temperature (2 m aboveground) (available at http://www.ecmwf.int as NC files with a spatial resolution of 0.25°) were used to analyze its relationship with NDVI variation from 2000 to 2017. Total precipitation and the average temperature of each year were calculated for analysis. For harmonization of datasets, temperature and precipitation, datasets were resampled to 250 m by using a bilinear resampling method [46,47] for analyzing its correlation with annual NDVI and growing season NDVI. MCD12Q1, MODIS Land Cover product of 2017 was acquired with a spatial resolution of 500 m [48]. Type 1 classification of MCD12Q1 was used in this research and classified the study area into 7 land cover classes that include barren, savannas, croplands, grasslands, water bodies, open shrub lands, and urban built-up areas (Figure 2b).

*2.3. Methods*

2.3.1. Linear Regression Model

A linear regression method was applied to analyze spatial and temporal fluctuations between annual NDVI, growing season NDVI and time using slope as an indicator of the direction and magnitude of time series trends, wherein a positive slope value indicates increasing vegetation trend, which indicates increase in vegetation coverage, while negative slope value can be an indicator of decreasing vegetation trend, which indicates decrease in vegetation coverage [16,49,50]. If the p-value of the regression is less than 0.05, then the trends are considered statistically significant. Pixels are considered static (no significant change) if the *p*-value is greater than 0.05. Pixels with *p*-value above 0.05 did not show enough sensitivity to detect the changing trends. NDVI (annual and growing season) trends are classified as No significant change ($p \geq 0.05$), significant increase (slope > 0 and for trend significance $p < 0.05$), significant decrease (slope < 0 and for trend significance $p < 0.05$).

2.3.2. NDVI-Climate Correlation and Driving Forces Analysis

Pearson correlation (R) and level of significance (p) between each climatic control (rainfall and air temperature) and NDVI (annual and growing season) over the 18 year study period were calculated respectively [51]. We believe that NDVI-climate control is significantly positive correlated if the R-value is positive and the p-value is lower than 0.05 between NDVI and climate control. To separate the influence of climate variability and other anthropogenic activities (multi-controls) on changing vegetation trends, the threshold segmentation method was used. If there is a significant positive correlation between NDVI and climate-control, then climate variability is considered as the main driving force of vegetation trend (increasing or decreasing). If there is no significant correlation then multi-controls are considered as the main driving forces of vegetation decreasing or increasing.

Drivers of changing trends in vegetation activity are classified into 4 categories (1) "Significant increase caused by climate variability" where during last 18 years climate variability-NDVI shows significantly positive correlation and NDVI increased significantly; (2) "Significant decrease caused by climate variability" regions where during last 18 years climate variability-NDVI shows significantly positive correlation and NDVI decreased significantly; (3) "significant vegetation increase caused by multi-controls" regions which shows significant increase but does not show a significant positive correlation. Multi-controls include the influence of ecological restoration projects, agricultural land-use,

etc., and (4) "anomalous decrease" regions, where decrease in vegetation is caused by population growth or other anthropogenic activities like urban sprawl. In such regions vegetation shows a significant decreasing trend but climate variability and NDVI are not significantly positively correlated.

### 2.3.3. Future Vegetation Trends: Hurst Exponent

The Hurst exponent is an effective method to describe the long term dependence of a time series and is widely used to estimate the persistence or anti-persistence of trends in a time series [52,53]. Hurst exponent quantifies a time series relative tendency to regress strongly to the cluster in a direction or to the mean [54]. An H value above 0.5 indicates long term positive autocorrelation within a time series. It means that a high value in the time series will possibly be followed by another high value and the future values will also tend to be high for a long time (persistent). An H value below 0.5 indicates long term switching between low and high values in the time series, which means a high value will possibly be followed by a low value and then again a high value so the future values are tend to be switching between low and high values for a long time in future. A totally uncorrelated time series can be indicated by an H value equal to 0.5 [55]. Assuming that there is long-term memory in our time series, the Hurst exponent is used. The range of H values is 0 to 1 [56,57]. For each pixel, the H value was calculated to test the persistence of future vegetation trends. Vegetation trends of the future can be classified into six types using vegetation trends and Hurst exponent (Table 1) [22].

**Table 1.** Trends of future vegetation based on vegetation dynamics (2000–2017) and H value.

| Sr. | Trend | H-Value | Description |
|-----|-------|---------|-------------|
| 1 | Positive trend | >0.5 | Positive Development |
| 2 | Positive trend | <0.5 | Anti-persistent positive development |
| 3 | Negative Trend | >0.5 | Negative Development |
| 4 | Negative Trend | <0.5 | Anti-persistent negative development |
| 5 | No Significant Trend | >0.5 | Stable and steady development |
| 6 | No Significant Trend | <0.5 | Undetermined development |

If in a pixel of vegetation, the trend is positive and the h value is greater than 0.5, then that pixel shows positive development and the trend is likely to continue. If the trend of vegetation activity is positive and h value is less than 0.5, then the pixel has anti-persistent positive development which indicates that the trend is not likely to continue. When a pixel has a negative trend and h value greater than 0.5, the negative trend of vegetation is expected to continue, and the overall development is negative. If the trend is negative and the h value is less than 0.5, then pixel shows an anti-persistent negative development after time series and the negative trend is unlikely to continue. If there is no significant trend and H value is higher than 0.5, then the pixel is stable. When vegetation trend is not significant, and H is less than 0.5, then there is an uncertain future vegetation trend after the time series and overall development is undetermined.

To estimate the auto-correlation properties of time series, Hurst [58] developed a "rescale range analysis". In our time series as a measure of long term memory, we will apply the Hurst exponent (H), using rescale range (R/S) analysis. The main steps are

- To divide the time series $\{ \xi (\tau) \}(\tau = 1, 2 \ldots, n)$ into sub-series x (t), and for each sub-series t = 1 $\ldots, \tau$.

- To define the mean sequence of the time series,

$$\langle \xi \rangle \tau = \frac{1}{\tau} \sum_{t=1}^{\tau} x(t), \ \ \tau = 1, 2, \ldots, n \tag{1}$$

- To calculate the cumulative deviation,

$$x\,(t,\tau) = \sum_{u=1}^{t} (\xi(u) - \langle\xi\rangle\tau),\ 1 \le t \le \tau \tag{2}$$

- To create the range of sequence,

$$R(\tau) = \max_{1\le t\le\tau} X(t,\tau) - \min_{1\le t\le\tau} (t,\tau), \tau = 1,2\ldots,n \tag{3}$$

- To create the standard deviation sequence,

$$S(\tau) = \sqrt{\frac{1}{\tau}\sum_{\tau=1}^{\tau} (\xi(t) - \langle\xi\rangle\tau)^2}, \tau = 1,2\ldots,n \tag{4}$$

- To rescale the range,

$$R(\tau)/S(\tau) = (C\tau)H \tag{5}$$

### 2.3.4. Validation

Randomly reordered test is the most commonly used technique to validate R/S analysis. The main steps are to reorder the original time series randomly and to compute the new the Hurst exponent. It was supposed that the randomly reordered time series was a random sequence, and according to the definition of indicator the new the Hurst exponent should be equal to 0.5. Thus, whether the H-value of newly generated time series was close to 0.5 would be taken as the criterion to detect the validity of R/S analysis [23].

## 3. Results

### 3.1. Vegetation Dynamic Trend and Driving Forces Analysis

The average annual NDVI was 0.21 and growing season NDVI was 0.22 for overall study area from 2000 to 2017, varying between land cover classes. Figure 4 illustrates the annual and growing season NDVI trends from 2000 to 2017. The slope of each pixel in the time series was calculated. The overall linear regression model (LRM) slope of annual NDVI and growing season NDVI for the whole study area was 0.0021 and 0.0023 per year, respectively. Positive trends in annual and growing season can be seen in the study area, which indicates major vegetation increase during the study period.

Results of the linear regression model revealed that over past 18 years, 86.71% (45.37% significant increase) of the study area showed an upward trend, 13.28% showed a downward trend (3.06% significant decrease) in annual NDVI; 82.7% (35.2% significant increase) of the study area showed an upward trend and 17.3% accounted for downward trends (3.3% significant decrease) in growing season NDVI (Figure 4). Furthermore, 51.55% and 61.5% area showed no significant trends in annual and growing season NDVI, respectively.

The highest significant increase in annual trend analysis was observed in croplands; that is, 65.61% of total cropland area, followed by grassland (51.09%), savannas (50.11%), urban (30.73%), barren (29.32%) and shrublands (29.00%), whereas urban area showed the highest significant decrease; that is, 28.56%, followed by grassland (7.64%), savannas (5.92), croplands (2.11%), barren (2.00%), and shrublands (1.24%).

During growing season, savannas showed the highest significant increase; that is, 46.8% of the total area of savannas, followed by croplands (46.2%), grassland (35.8%), urban (29.22%), barren (29%), and shrublands (20.8%). Contrasting trends to this, the highest significant decrease was observed in urban areas that accounted for 19.5% of the total urban area, followed by grassland (7.2%), savannas (3.5%), croplands (3.4%), barren (1.6%), and shrublands (1.2%).

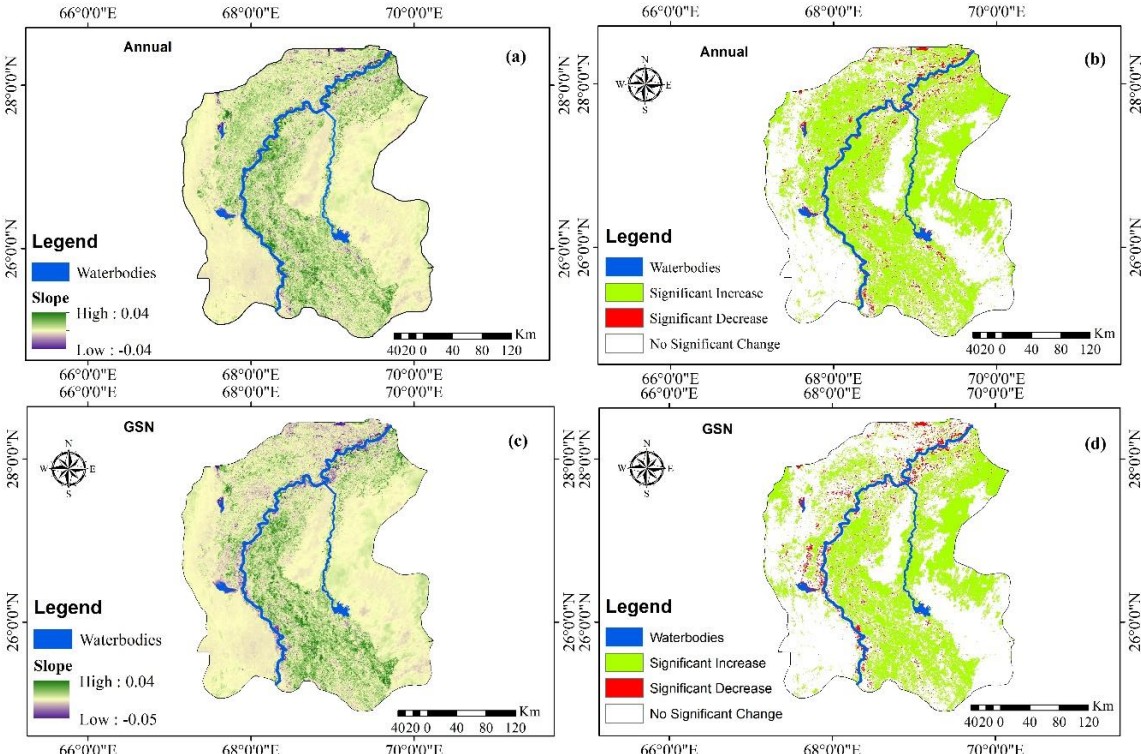

**Figure 4.** (**a**) Annual NDVI linear regression model (LRM) slope from 2000 to 2017, (**b**) annual distribution of significant increasing and decreasing trends ($p < 0.05$), (**c**) Growing Season NDVI LRM slope from 2000 to 2017 (**d**), Growing Season distribution of significant increasing and decreasing trends ($p < 0.05$).

The vegetation dynamics process is very complex, so if the relative contributions of climate and anthropogenic causes remain unclear, then the driving forces of vegetation dynamic activity will remain obscure. Figure 5a shows the Annual NDVI-Rainfall correlation significance. The total area shows a positive relationship between rainfall and Annual NDVI of 78.34%. Figure 5b shows Annual NDVI-Temperature correlation significance; the area shows a positive correlation between temperature and Annual NDVI of 23.16%.

Out of the 45.37% significant increase in annual vegetation only 4.97% was caused by climate variability, the other 40.40% significant increase was caused by multi-controls (ecological restoration projects, and agricultural land-use), whereas out of the 3.06% significant decrease in annual vegetation, only 0.27% was caused by climate variability, and 2.79% decrease was anomalous. Increasing trends of vegetation activity caused by multi-control dominates all land use classes: cropland at 97.02%, followed by grassland at 95.95%, savannas at 93.50%, urban at 92.27%, open shrublands at 73.17%, and barren at 72.20%. Significant vegetation decrease in all land use classes was mainly anomalous; that is, above 90% in all land use classes other than barren lands; that is, 84.10%.

Figure 6a shows the Growing Season NDVI and Rainfall (GSN-Rainfall) correlation significance. The total area shows a positive relationship between rainfall and growing season NDVI of 79.14%. Figure 6b shows GSN-Temperature correlation significance. The area shows a positive correlation between temperature and growing season NDVI of 58.63%. When comparing Figures 2b and 6a, it can be observed that high correlation pixels are mainly distributed over the east and west part of the study area, whereas land use is mainly dominated by barren land. The reason for this result is that abundant water is not available there, which enhances the dependence of vegetation on precipitation. Moreover, the western part of the study area is covered by the Thar dessert, which is the only fertile desert in the world; fertile soil helps vegetation to grow. Due to the hot and arid geographical environment, an

increase in precipitation can increase the vegetation growth in these areas by limiting the effects of low water availability.

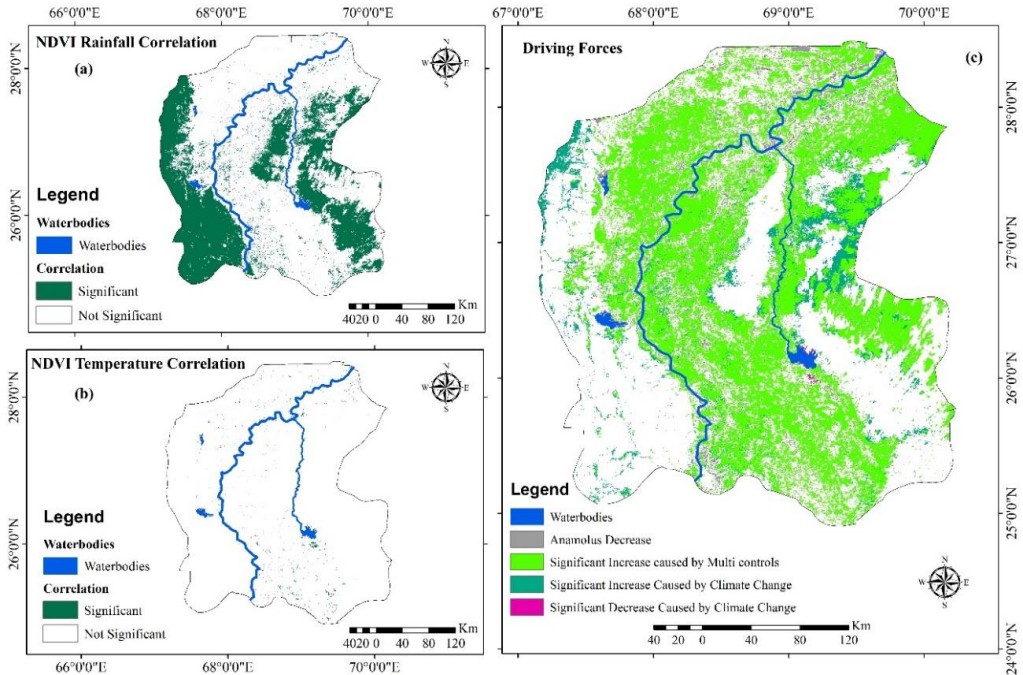

**Figure 5.** (**a**) Spatial pattern of Annual NDVI and rainfall correlation significance, (**b**) spatial pattern of Annual NDVI and temperature correlation significance, and (**c**) driving forces distribution of annual vegetation dynamics.

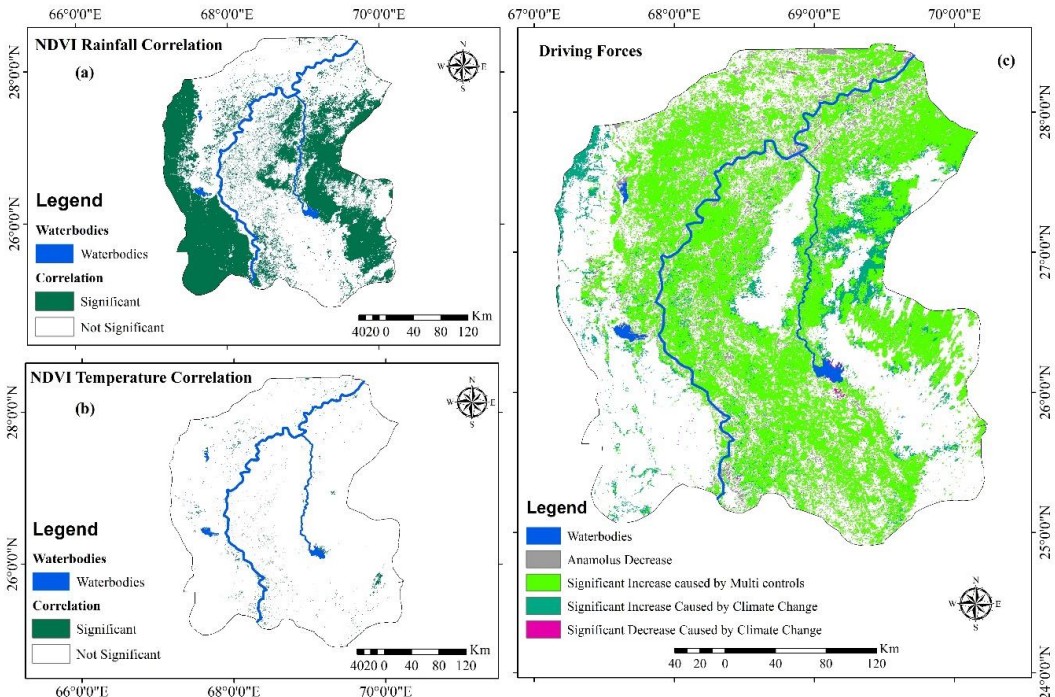

**Figure 6.** (**a**) Spatial pattern of GSN and rainfall correlation significance, (**b**) spatial pattern of GSN and temperature correlation significance, and (**c**) driving forces distribution of vegetation dynamics.

Out of the 35.2% significant increase in vegetation, only 9.5% was caused by climate variability, and the other 25.68% significant increase in vegetation was caused by multi-controls (ecological restoration

projects, and agricultural land-use), whereas out of the 3.39% significant decrease, only 0.09% was caused by climate variability and 3.30% decrease was anomalous. Other than barren lands, among all classes significant increase was mainly caused by multi-controls. Only barren lands (51.29%) showed an increasing trend because of climate variability, and the 48.60% increase was caused by multi-control. Increasing trends caused by multi-control dominates all other land use classes: grassland at 88.78%, followed by cropland at 86.83%, savannas at 83.86%, urban at 79.90%, and open shrublands at 53.17%, whereas the highest significant decrease in vegetation in barren lands was anomalous. Significant vegetation decrease in all land use classes was mainly anomalous; that is, above 90% in all land use classes.

### 3.2. Trends of Future Vegetation based on H-value and Vegetation Dynamics

The overall H value of annual Hurst was 0.65 and H ranged from 0.23 to 0.75 (Figure 7a), whereas in growing season overall Hurst was 0.61 and H ranged from 0.16 to 0.71 (Figure 8a). The annual Hurst value 0.65, and growing season Hurst value 0.61, indicates that there is an overall high consistency in the future vegetation trends in arid regions of Sindh province. This indicates that the vegetation trends in the future will be similar to those of the past 18 years. Based on H-values, 95.98% of the area showed consistency in annual vegetation trends, whereas 4.02% area showed anti-persistent annual vegetation trends. On the other hand, 76.62% of the area showed persistent growing season vegetation trends, while only 23.37% of the whole study area showed inconsistency, which can be observed in the south-east and north-west part of the study area. The north-western part of the study area that is showing inconsistency is prone to droughts, and this inconsistency could be due to extreme drought conditions in that region. According to land cover class results, all classes showed an H value greater than 0.5 during both annual and growing season, which indicates the consistency of trends in every land cover class. Barren land had the highest mean H-value (annual 0.65 and seasonal 0.57), followed by urban (annual 0.64 and seasonal 0.56), savanna and grassland (annual 0.63 and seasonal 0.55), cropland (annual 0.62 and seasonal 0.54), and open shrublands (annual 0.61 and seasonal 0.51). Barren land had the highest H value, which could be due to low anthropogenic activities in that area.

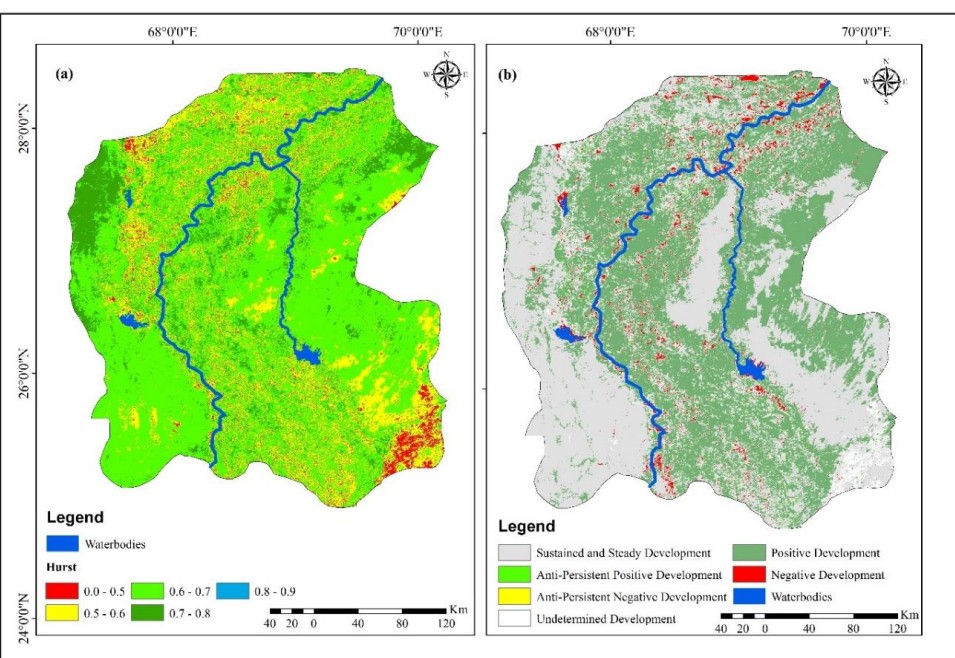

**Figure 7.** (**a**) Map of the Annual Hurst Exponent of the study area; (**b**) Annual Future vegetation trends based on H-value and vegetation dynamics (2000–2017).

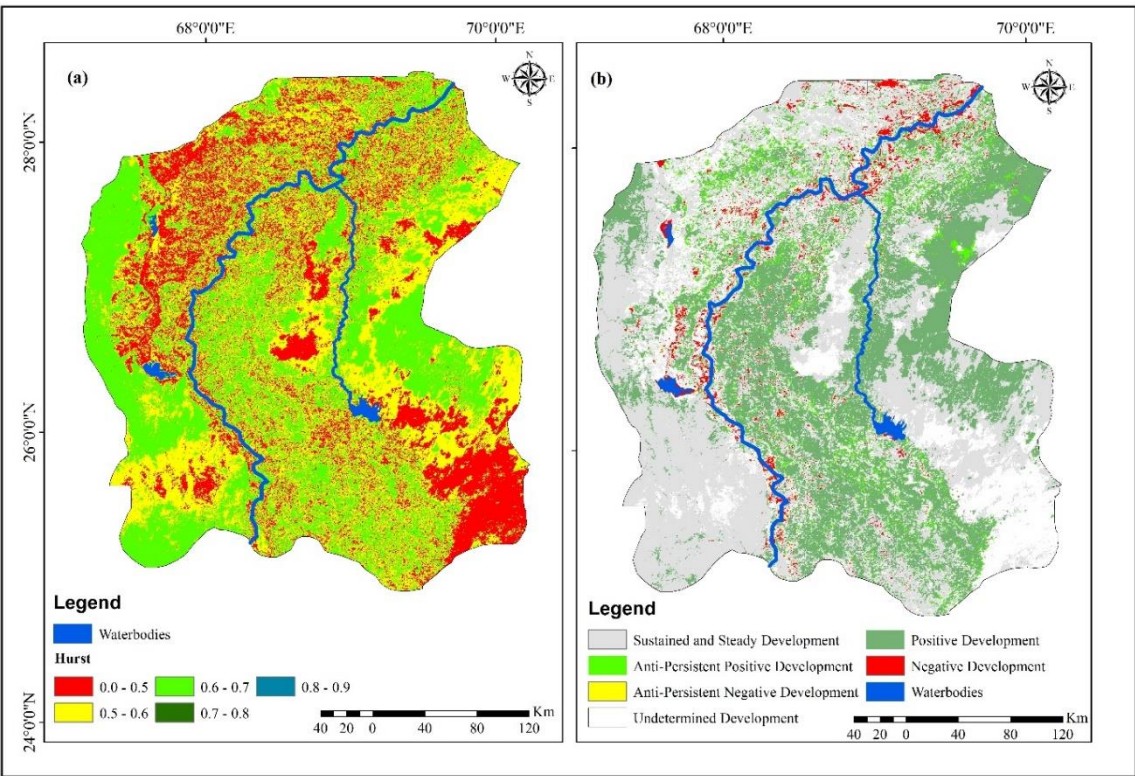

**Figure 8.** (**a**) Map of Growing Season Hurst Exponent of the study area; (**b**) Growing Season Future vegetation trends based on H-value and vegetation dynamics (2000–2017).

To indicate the trends of future vegetation, H value and vegetation dynamics were combined into six different classes (Figures 7b and 8b). The annual trends of future vegetation show sustained and steady development (48.45%), followed by Positive development (44.49%), Undetermined (3.05%), Negative Development (2.09%), Anti-Persistent positive development (0.92%), Anti-Persistent negative development (0.09%). Likewise, the growing season trends of future vegetation show sustained and steady development (42.80%), followed by Positive development (30.77%), Undetermined (18.56%), Anti-Persistent positive development (4.46%), Negative Development (3.04%), and Anti-Persistent negative development (0.35%).

Results revealed that among all land cover classes, cropland (annual 63. 76% and growing season 37.77%) and grassland (annual 49.80% and growing season 30.01%) have the highest positive development in its total area, which indicates that positive trends (vegetation increase) are likely to continue in cropland and grasslands, whereas the highest annual negative development can be observed in urban (28.01%) and grasslands (7.32%), which indicates that the negative trend is likely to continue, while in growing season grassland show the highest negative development (6.5%) followed by urban areas (3.49%).

Both climate variability and multi-control have an impact on future vegetation trends. Figure 9 illustrates that during the annual study period out of total significant increase, 89.03% was caused by multi-control out of which 83.86% is likely to continue in future, which indicates that the 94.19% significant increase was caused by multi-control shows persistent trends. Moreover, 10.96% of the total significant increase was caused by climate variability, out of which 10.24% showed consistency that 93.46% of the significant restoration was caused by climate variability. Contrary to this, during the growing season, out of total significant increase, 72.89% was caused by multi-control out of which 61.31% was likely to continue in future, which indicates that 84.11% of the significant increase was caused by multi-control and shows persistent trends. Climate variability caused 27.11% of total significant increase, out of which 25.99% showed consistency that was 95.93% of the significant

restoration caused by climate variability. Climate variability has a more considerable impact on future vegetation than multi-control during growing season with local weather conditions (i.e., rainfall and temperature) that permit normal plant growth.

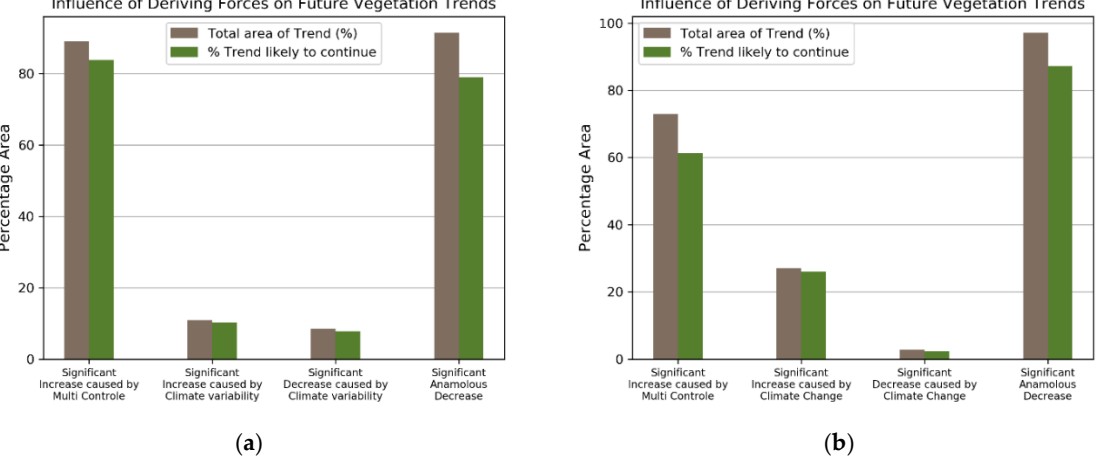

**Figure 9.** Influence of driving forces on future vegetation trends, (**a**) Annual, and (**b**) Growing Season.

On the other hand, during the annual study period, out of total significant decrease in vegetation, 91.48% was significant anomalous decrease, out of which 79.04% is likely to continue in future, which indicates that the 86.40% significant anomalous decrease shows persistent trends. Climate variability caused 8.51% of total significant decrease out of which 7.72% showed consistency that was a 90.69% significant decrease caused by climate variability. Contrary to this, during growing season out of total significant decrease in vegetation, 97.15% was a significant anomalous decrease out of which 87.31% is likely to continue in future, which indicates that the 89.87% significant anomalous decrease shows persistent trends. Furthermore, 2.84% of the total significant decrease was caused by climate variability, out of which 2.25% shows consistency that is 79.41% of significant decrease caused by climate variability.

## 4. Discussion

Pakistan committed itself towards attaining the millennium development goals with other UN member states, in the year 2000 [59]. To meet millennium development goals, it is essential to address desertification [60]. Therefore, it is essential to quantify the increasing and decreasing vegetation trends. Climatic variations and anthropogenic activities are the two key driving forces of degradation trends [61]. Several studies have suggested that changing climate is one of the key driving forces of vegetation trends [62,63]. Our results demonstrate that the role of climate variability in vegetation dynamics is limited confirming the results of [22] and [13]. Another recent study [64] has reported a more significant role of the direct human factor in the greening of the earth than climatic factors, and our results support their conclusion.

The linear regression model, Pearson's correlation, and the Hurst exponent are used to investigate vegetation trends, driving forces, and their persistence. Based on these results, a future vegetation trend is generated and impact of climatic and anthropogenic activities on future vegetation trends was calculated. Though the use of linear regression model to study vegetation dynamics has been criticized as being unable to reflect nonlinear characteristics [65] and where mutation is smoothed in the regression, it is still widely used method because of its simplicity and robustness, in large-scale vegetation dynamics studies [66]. The study found an overall increasing trend for both the annual and growing season. Quantifying the relative contributions of climate and anthropogenic factors is important in order to identify the driving forces behind vegetation dynamics. Results suggested that the positive relationship between NDVI and rainfall is much stronger than the relationship

between NDVI and temperature during both annual (Figure 5a,b) and growing season (Figure 6a,b). Semi-arid regions are extremely sensitive to precipitation variability and its timing [67]. Even a small increase in precipitation can cause large increase in plant available soil water and subsequent plant growth [68]. Results of this study revealed that during growing season, savannas show the highest significant restoration, it can be the result of an increase in precipitation intensity during growing season. A recent study [69] reported that the response of savannas to a little increase in precipitation is very quick. Results of another study [70] have also confirmed the positive relationship between rainfall and vegetation activity in savannas. It can be observed from Figures 5c and 6c that the relationship between vegetation and multi-control is very strong as compared to the climate variability alone during both periods.

Among all land use and land cover classes significant increase in vegetation was derived by multi-controls during both annual and growing season. Interestingly, only in barren land during growing season was the significant increase driven by climate variability. The significant increase in grassland and cropland since 2000, due to multi-control, is mainly attributed to direct factors like government policies or better agriculture practice (multiple cropping, better seeds, use of fertilizers, pest control, etc.). The influence of climate variability in barren land is strong because of low anthropogenic activity in the area (no cultivation or restoration activities), and the area of barren land increases or decreases with changing climate (increase or decrease in rainfall) [71].

Results (Figures 7a and 8a) revealed that in the vegetation trends of the study area there is an overall high consistency during both annual and growing. However, the consistency in annual trends is much higher than the growing season trends. 95.98% area shows persistent annual vegetation trends, whereas 4.02% area shows anti-persistent annual vegetation trends based on the Hurst value. The study area shows a high consistency in vegetation trends of 76.62%, while only 2.37% of the study area shows the inconsistency in growing season vegetation dynamics, which is mainly distributed in shrub and barren lands. It is convincing that the main reason for the inconsistency is low water availability (away from river and major canal), and the area is also prone to droughts. Sadiq [72] has investigated extreme climatic events including flood zones, drought trends, and severity in their study. The results of their study revealed that a number of districts in Sindh province especially in north-western part of Sindh (Jacobabad, Shikarpur, Larkana, Nowshehro Feroz, and Dadu etc.) are most susceptible to drought. In terms of the type of land use, barren land located primarily in the eastern part of the study area shows high consistency, because it is comprised of sandy and less fertile soil, so it is the least affected area by positive and consistent human activates i.e., agriculture and restoration projects [23]. Upward trends of vegetation driven by climate variability are more likely to continue, whereas overall decreasing trends under climate influence is already very low and climate variability has caused increase in vegetation in such areas where less anthropogenic activity has observed.

### 4.1. Validation of R/S Analysis

Hurst exponent is not very commonly used for time series analysis to study vegetation dynamics. Therefore, it is essential to ensure the validity of the estimated Hurst exponent as to whether it is suitable or not. As suggested by [23], a randomly reordered test is the most commonly used method to identify the validity of R/S analysis. The original time series was reordered randomly, and the Hurst exponent of the newly generated time series was estimated. According to the definition of the indicator [58], the new Hurst exponent should be close to 0.5. As shown in the Figure 7, the Hurst exponent of the newly generated time series is 0.64 for annual and 0.56 for growing season, which is above 0.5, and hence, validates our results.

Moreover, from the scatterplot of annual and growing season between the Hurst exponents of original and randomly reordered time series as shown in the Figure 10, it can be observed that the majority of the Hurst exponents of the original time series are above 0.5 and spread towards 0.7. Likewise, the Hurst exponent of the newly generated time series are close to 0.5 and are smaller than

the original. Furthermore, [73] have also discussed the excellent applicability of the Hurst index in vegetation change persistence analysis in arid and semi-arid regions.

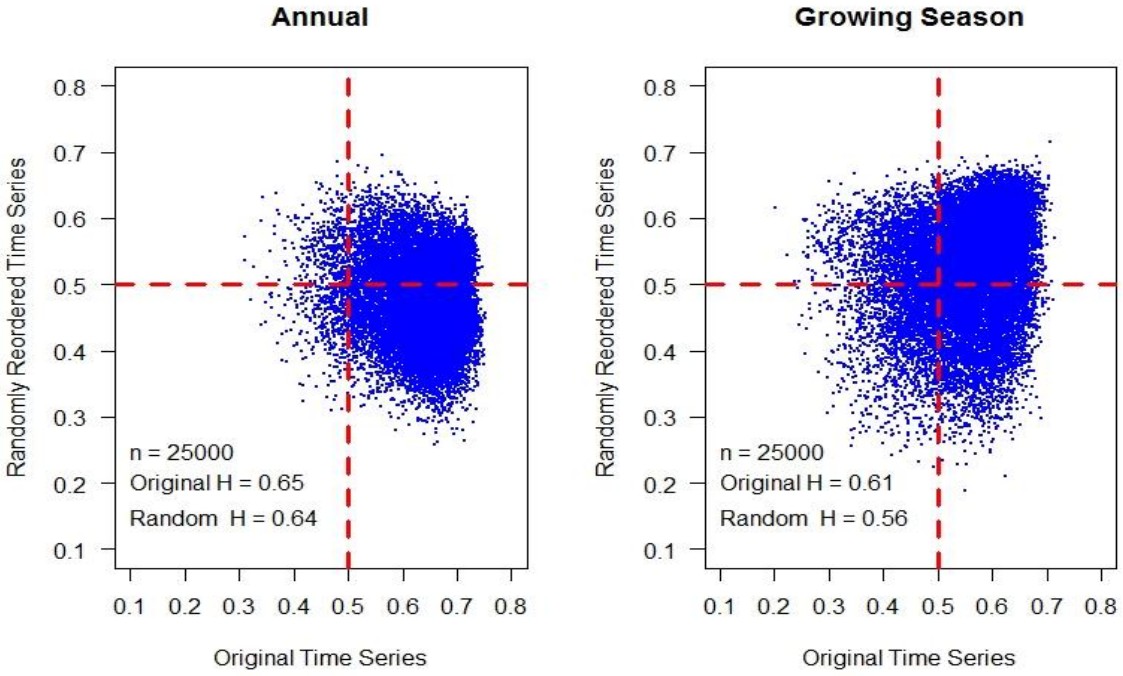

**Figure 10.** Scatterplot of annual and growing season between the Hurst's exponents of original and randomly reordered time series.

*4.2. Limitations and Future Research Directions*

In this study, authors have used MOD13Q1 MODIS NDVI Product (2000 to 2017) with a spatial resolution of $250 \times 250$ m to study vegetation dynamics from 2000 to 2018. Better temporal and spatial resolution datasets should be used in the future to avoid uncertainties caused by cloud cover or low spatial resolution. Moreover, further research should be done to find out how long the trend will continue in the future based on ongoing policies. Despite of the fact that NDVI is being largely used from local to global scale in vegetation dynamics studies future studies must consider problems related to NDVI including noisy canopy background signal and saturation. We have explored both climatic and anthropogenic drivers of vegetation activity changing trends but did not statistically distinguish the impact of ecological restoration projects (large scale plantation) from other anthropogenic activities (agriculture), which will further aid in estimating the success of restoration projects. Solar radiation should also be considered in future studies as some researchers proposed that solar radiation also affects vegetation dynamics.

## 5. Conclusions

This study examined spatial and temporal changes of vegetation coverage for annual and growing season over the arid and semi-arid regions of Pakistan from 2000 to 2017. Satellite data has shown improvement in overall vegetation conditions in the arid and semi-arid region of Sindh during the study period, and decline can only be seen in a small area. Based on the above analysis following conclusions could be drawn:

(1) Overall arid and the semi-arid regions have been shown to be characterized by improved vegetation conditions from 2000 to 2017 with significant increase in 45.37% and significant decrease in 3.06% area during annual, respectively, while 35.2% increase and 3.3% decrease during growing season, respectively, whereas most of the area has shown no significant trends.

(2)  Most of the considerable decline happening in the study area is anomalous during both annual and growing season, and it is in happening in the barren land. Significant increase over most of the study area during both seasons due to multi-control, only during growing season in barren lands, and 51.29% area shows vegetation increase because of climate variability.

(3)  There is an overall high consistency in future vegetation trends in arid regions of Sindh province. Stable and steady development region (48.45% during annual and 42.80% during growing season) dominates the future vegetation trends and is followed by positive development.

(4)  Croplands and grasslands showed the highest positive development in its total area that is 63.76% and 49.38% during annual and 37.77% and 30.01% during growing season, respectively, whereas the highest negative development can be observed in urban 28.01% and grassland 7.32% during annual, while grasslands 6.5% and urban areas 3.49% during growing season.

(5)  Multi-controls have slightly larger impact on future vegetation upward trend than climate variability during annual trends, but the difference is not very large. Significant increase caused by multi-control and climate variability that is likely to continue is found at 94.19% and 93.46% respectively.

(6)  Climate variability has a more considerable impact on future vegetation upward trends than multi-controls during growing season. The 95.93% out of total increase under the influence of climate variability is likely to continue, and the 84.11% out of total increase under the influence of multi-controls is likely to continue during growing season.

(7)  Climate variability controls future annual decreasing vegetation trends with 90.69% of overall decrease caused by climate variability is likely to continue in the future. Anomalous decrease has more control over future growing season decreasing trends. The 89.87% of overall decrease caused by anthropogenic factors (urbanization and population growth) is likely to continue in the future.

(8)  The study area is characterized by an improved vegetation conditions from 2000 to 2017 with significant restoration, and similar trends were estimated for the future.

The results of the present study will enrich our knowledge about the influence of climatic and anthropogenic factors of vegetation dynamics on future vegetation trends in arid and semi-arid regions. Future vegetation increasing trends shows a good response towards human efforts, but at the same time, anthropogenic factors are dominating factors of future downward vegetation trend as well. Further efforts should be made to limit future downward trends and to improve restoration activates.

**Author Contributions:** Conceptualization, B.B. and S.N.; methodology, B.B.; software, B.B.; validation, B.B. and M.Z.J.; formal analysis, B.B and K.J.; investigation, B.B.; resources, B.B.; data curation, B.B.; writing—original draft preparation, B.B. and H.A.; writing—review and editing, B.B, X.B., and F.M.; visualization, B.B.; supervision, C.C.; and funding acquisition, C.C. All authors have read and agreed to the published version of the manuscript.

**Funding:** This research was financially supported by the National Key Research and Development Program of China "Research of Key Technologies for Monitoring Forest Plantation Resources" (No. 2017YFD0600900) project.

**Acknowledgments:** Six authors, Barjeece Bashir, Mehdi Zamani Joharestani, Shahid Naeem, Xie Bo, Kashif Jamal, and Faisal Mumtaz acknowledge University of Chinese Academy of Sciences (UCAS). Barjeece Bashir acknowledges China Scholarship Council for awarding Chinese Government Scholarship and support to carry out this research.

**Conflicts of Interest:** The authors declare no conflict of interest.

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
