# Peer review of "Spatio-Temporal Vegetation Dynamic and Persistence under Climatic and Anthropogenic Factors"

_remotesensing, doi:10.3390/rs12162612_

Round 1

Reviewer 1 Report

This is an interesting work, well-written and well-structured, and with intresting findings. 

Only, one comment in order to improve the value of the findings. Authors should try to more document (ecologically based), the interesting finding that the study area is characterized by an improved vegetation conditions from 2000 to 2017 with significant restoration, and similar trends were estimated for the future.

Author Response

Thank you for the positive feedback. Please see the attachment. 

Reviewer 2 Report

The manuscript shows an analysis of spatial and temporal changes of vegetation coverage for the annual and growing season over Pakistan from 2000 to 2017. In general, analyzes and techniques used are very robust and interesting. However, the discussions and conclusions about the impact of Climate change on vegetation restoration are not supported by the analyzes and data considered in this study. A time series long enough to assess the scale of climate change was not used. Therefore, I recommend that either the authors change the terminology throughout the entire manuscript or consider including a longer period for analysis.

Also, I consider that a little more explanation about the physics of processes is missing. The manuscript has the potential to improve substantially.

Line 75: What is the definition of harsh drought used in this work? Physical, social, environmental, etc.?

Line 106: Include the acronym IGBP.

Figure 2. The difference between figures 2a and 2b is very subtle, almost imperceptible, in that sense, I suggest including a map of the difference between annual NDVI and the growing season NDVI.

Lines 149-151: Rather than mentioned the name of the used software, I find it more useful to describe what exactly the tool does and how it does it.

Line 156: It is not reasonable to evaluate the climate change scale with time-series of only 18 years, please justify.

Figure 3 – The same issue mentioned in figure 2. The difference between figures 3a,b and 3dc are very subtle, almost imperceptible, in that sense, I suggest including maps of the difference.

Please consider improving the quality of the figures, fonts, scales, etc.

Line 228: About the sentence: "During growing season savannas show the highest significant restoration", this is an expected result due to the rainy season that normally occurs, isn't it?

Lines 248 – 250: Figure 1 or 5? I suggest showing on the maps only pixels with correlation with statistical significance. The same for Figure 6.

Lines 286 – 288 and 361-362: The authors mention severe droughts, but this has never been evaluated. I suggest showing some drought indicators to support the statements. For example, an SPI time series could be used.

Line 315: I suggest the authors review the concept of climate change used throughout the manuscript. It is not correct to assess climate change with a short time scale. This should be done with at least 30 years of data.

Author Response

Thank you for giving us the opportunity to submit a revised draft of the manuscript. We appreciate the time and effort that you dedicated to providing feedback on our manuscript and are grateful for the insightful comments on and valuable improvements to our paper.We have incorporated most of the suggestions made by the reviewer. Those changes are highlighted within the manuscript. Please see the attached file, for a point-by-point response to the reviewer. Thank you.

Reviewer 3 Report

This manuscript is generally well written but it can be improved. Introduction does not contain information directly related to NDVI though NDVI is the key parameter used in this study. The applications and issues of NDVI can be included briefly in Introduction and expanded in Discussion. A recent NDVI review paper by Huang et al. (2020) (A commentary review on the use of normalized difference vegetation index (NDVI) in the era of popular remote sensing. Journal of Forestry Research. https://link.springer.com/search?query=&search-within=Journal&package=openaccessarticles&facet-journal-id=11676) provides inspirations about NDVI issues, particularly its saturation problem, which may affect the analysis results in this paper. 

Author Response

We are very grateful for the reviews provided by the reviewer of this manuscript. Please see the attached file, for a point-by-point response to the reviewer.

Reviewer 4 Report

This manuscript deals with a time-series analysis of NDVI data in Pakistan. The text is correctly organized, the figures and tables are all necessary and clear, and the quality of presentation is good. The English requires moderate edition, in most cases because probably earlier versions became inherited in the current one, resulting in a few awkward phrases.

The title describes well the authors' aims at making this study. With respect to finding vegetation dynamics by linear regression models, the work itself is correctly done. But the authors do not seem to have accounted for the extense literature published on vegetation trend analysis. As a result, there are important conceptual gaps preventing a correct interpretation of the results. I found particularly difficult to accept their denominations 'restoration' and 'degradation' for an analysis not involving examining regression residuals or any alternative to this.

Concerning the part of persistence found by the Hurst exponent, I agree with the authors in that this technique has not been often used to analyze vegetation time-series, and found very provoking (in the best sense) their combination of H-values and trends. However, for this part to be really important, the conceptual frame suporting the given rules should be better justified.

My overall impression is that the technical component of this study is well carried out, and I found no relevant flaws in the methods or their implementation. Nevertheless, the conceptual framework, assumptions and resulting interpretations should be substantially improved. For that reason, I have focused my observations below on these aspects.

The above comments justify my recommendation to accept this paper after a 'minor' revision, which should be however rather extensive. I hope this encourages the authors to improve this interesting study.

Author Response

Thank you for the opportunity to revise our manuscript. The suggestions offered by the reviewers have been immensely helpful, and we also appreciate your insightful comments on a number of aspects of the paper. We have incorporated most of the suggestions made by the reviewer. Those changes are highlighted within the manuscript. Please see the attached file, for a point-by-point response to the reviewer. Thank you. 

Round 2

Reviewer 2 Report

The authors have addressed all suggestions from me and the manuscript is now significantly improved and acceptable for publication.

This manuscript is a resubmission of an earlier submission. The following is a list of the peer review reports and author responses from that submission.